# Unraveling the crucial role of trace oxygen in organic semiconductors

Yinan Huang[1,2], Kunjie Wu[3], Yajing Sun[2], Yongxu Hu[2], Zhongwu Wang [ID][2], Liqian Yuan[2], Shuguang Wang[2], Deyang Ji [ID][2], Xiaotao Zhang [ID][2], Huanli Dong [ID][4], Zhongmiao Gong[3], Zhiyun Li[3], Xuefei Weng[3], Rong Huang[3], Yi Cui [ID][3], Xiaosong Chen [ID][2] ✉, Liqiang Li [ID][1,2,5] ✉ & Wenping Hu [ID][1,2,5,6]

Optoelectronic properties of semiconductors are significantly modified by impurities at trace level. Oxygen, a prevalent impurity in organic semiconductors (OSCs), has long been considered charge-carrier traps, leading to mobility degradation and stability problems. However, this understanding relies on the conventional deoxygenation methods, by which oxygen residues in OSCs are inevitable. It implies that the current understanding is questionable. Here, we develop a non-destructive deoxygenation method (i.e., de-doping) for OSCs by a soft plasma treatment, and thus reveal that trace oxygen significantly pre-empties the donor-like traps in OSCs, which is the origin of p-type characteristics exhibited by the majority of these materials. This insight is completely opposite to the previously reported carrier trapping and can clarify some previously unexplained organic electronics phenomena. Furthermore, the de-doping results in the disappearance of p-type behaviors and significant increase of n-type properties, while re-doping (under light irradiation in $O_2$) can controllably reverse the process. Benefiting from this, the key electronic characteristics (e.g., polarity, conductivity, threshold voltage, and mobility) can be precisely modulated in a nondestructive way, expanding the explorable property space for all known OSC materials.

Organic semiconductors (OSCs) are promising candidates for the next generation of optoelectronic devices due to their low-cost, mechanical flexibility and large-area fabrication ability[1–3]. The weak molecular interaction and limited orbit overlap of OSCs make them susceptible to extrinsic defects and impurities[4–9]. Consequently, the observed optoelectronic property of most OSCs is generally a result of the interactions of their intrinsic nature with the extrinsic factors, in which some seemingly insignificant ones, even at ultralow levels (i.e., trace-level), play dominated roles[4,10–14]. This has led to a number of misunderstandings and challenges that have plagued scientific technological progress, and recently, some of them have been clarified after their origins were revealed, e.g., severe inhibition of silanol groups for *n*-type conduction[14], vital function of isomers for organic luminescence[12], and $H_2O$-induced pervasive charge trapping[4,13].

Among the many involuntary introduced extrinsic factors, the ubiquitous oxygen is the most prevalent impurity in OSCs, and numerous properties of materials and devices are closely related to it. Oxygen has long been considered a charge-carrier

[1]Joint School of National University of Singapore and Tianjin University, International Campus of Tianjin University, Fuzhou, Fujian 350207, China. [2]Tianjin Key Laboratory of Molecular Optoelectronic Sciences, Department of Chemistry, Institute of Molecular Aggregation Science, School of Science, Tianjin University, Tianjin 300072, China. [3]Suzhou Institute of Nano-Tech and Nano-Bionics (SINANO), Chinese Academy of Sciences, Suzhou, Jiangsu 215123, China. [4]National Research Center for Molecular Sciences, Institute of Chemistry, Chinese Academy of Sciences, Beijing 100190, China. [5]Haihe Laboratory of Sustainable Chemical Transformations, Tianjin 300192, China. [6]Collaborative Innovation Center of Chemical Science and Engineering, Tianjin 300072, China. ✉e-mail: xschen2019@tju.edu.cn; lilq@tju.edu.cn

trap in OSCs, leading to mobility degradation and stability problems[15–22]. This understanding experimentally relies on the conventional deoxygenation methods[16,23–26], e.g., annealing and sublimation. However, there is no doubt that trace oxygen residues are unavoidable in these processes. Therefore, the current understanding of oxygen's role in OSCs is questionable, and highly necessary to be re-investigated, which requires an effective deoxygenation method to complete remove the oxygen in OSCs that has not been developed until now.

Here, we find that trace amount of oxygen (-10$^{15}$ cm$^{-3}$) is prevalently inherent in a wide range of OCSs, even after severe purification. To unravel the underlying impact of the trace oxygen, we develop an uncommon deoxygenation method (i.e., de-doping) by an elaborately-controlled soft plasma treatment, and find that oxygen exhibits a very different role at trace levels than previously reported, i.e., trace oxygen significantly pre-empties the donor-like traps in OSCs rather than capturing charge carriers, which is the origin of the *p*-type characteristics exhibited by the majority of these materials. Furthermore, de-doping results in the complete disappearance of *p*-type behaviors of OSCs and over tenfold increase of their electron mobility (*n*-type property), while re-doping (under light irradiation in O$_2$) can controllably reverse the process. Benefiting from this, the key electronic characteristics (e.g., polarity, conductivity, threshold voltage ($V_T$), and mobility) can be precisely modulated in a non-destructive way, expanding the explorable property space for all known OSC materials. This work not only offers a new perspective to re-understand the intrinsic properties and perplexing observations of OCSs, but also provides a new way to modulate the charge transport in OSCs.

## Results and discussion

### Oxygen is prevalent inherent in OCSs

Firstly, a classic *p*-type OSC, dinaphtho[2,3-b:2′,3′-f]thieno[3,2-b]thiophene (DNTT) is selected as the candidate to investigate the latent trace oxygen in OSCs and devices. The measurements of photoelectron spectroscopy (XPS) suggest the significant decrease in oxygen concentration for freshly prepared DNTT films (materials are purified for three times before use) (Supplementary Section 1). However, the result of time of flight secondary ion mass spectrometry (TOF-SIMS) shows that trace oxygen still exists in the fresh single crystal by physical vapor transport (Fig. 1a). The valence photoemission and secondary electron spectra extracted from ultraviolet photoelectron spectroscopy (UPS) measurement of the freshly prepared DNTT films clearly show that the Fermi level ($E_F$) is close to HOMO rather than in the center of energy gap (Fig. 1b). When electron paramagnetic resonance spectroscopy (EPR) is used to track unpaired electrons, the samples exhibit an identifiable EPR signal (Fig. 1c) with $g = 2.0032$ (close to the free electron $g_e \approx 2.0023$). The temperature dependence of the EPR signal shows that the magnetic susceptibility ($X$) obeys the Curie law (Supplementary Section 2)[27]. Consequently, the signal is assigned to the DNTT radical cation with $S = 1/2$ spins in solid[28]. Meanwhile, when 5,5-dimethyl-1-pyrroline N-oxide (DMPO), a radical trapping agent, is used, the characteristic EPR signal of the adduct DMPO-•OOH[29] of DMPO with superoxide anion (O$_2^-$) is detected.

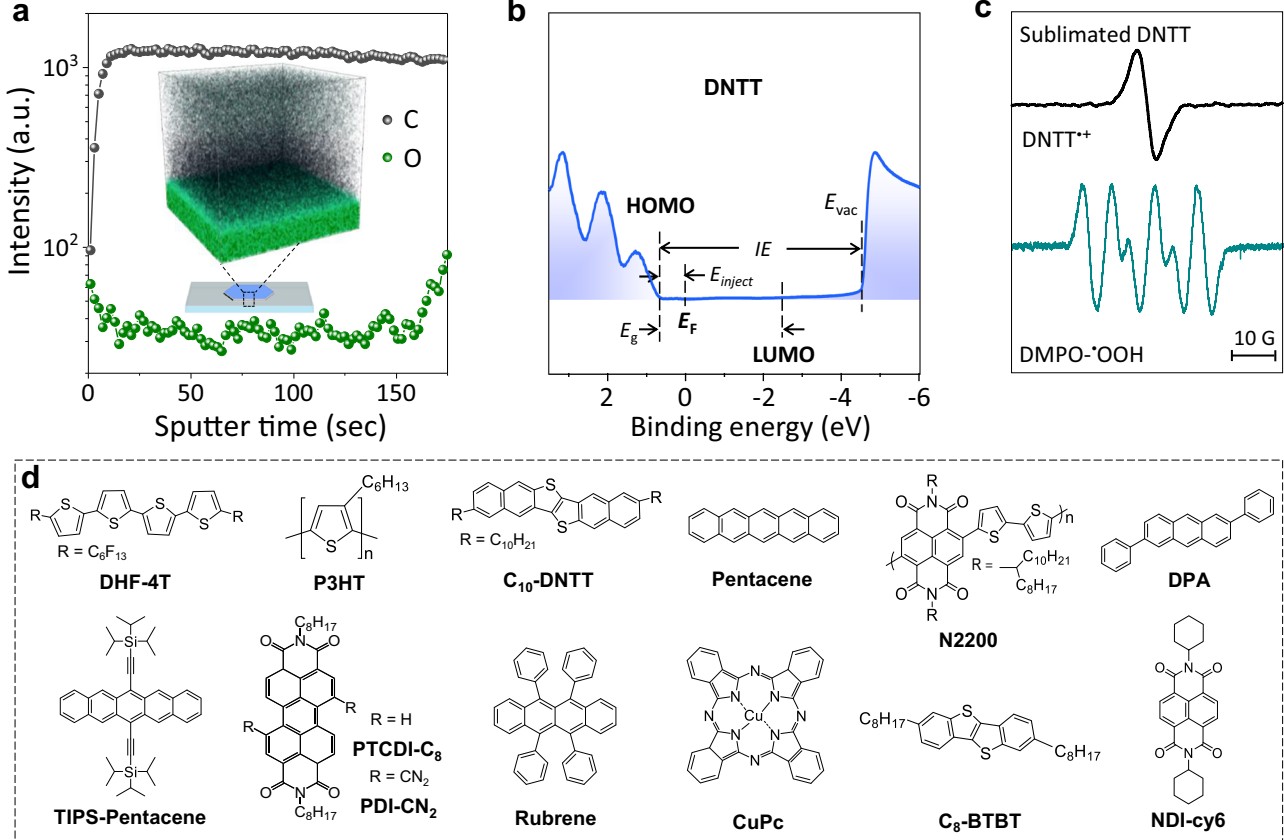

**Fig. 1 | Oxygen doping chemistry in OSCs and devices. a** Intensity of the TOF-SIMS data for the C and O elements of the fresh DNTT single crystal. Inset is the 3D reconstruction of the depth profile. **b** Valence photoemission spectrum and secondary electron spectrum extracted from the UPS measurement of the fresh DNTT film. Ionization energy (*IE*), vacuum level ($E_{vac}$), and injection barrier ($E_{inject}$) are exacted from the UPS result, and the energy gap ($E_g$) is obtained from the UV-vis absorption spectrum. The energy position of the HOMO, the LUMO, and the $E_F$ are thus determined. **c** The EPR signals of organic radical cations (ORCs) and DMPO-•OOH (the adduct of DMPO with superoxide anions (O$_2^-$)) in fresh DNTT film, suggesting the innate oxygen doping in OSCs. **d** Different OSCs were used for investigation in this work.

The results solidly confirm the innate and stable existence of oxygen dopants in OSCs, i.e., the geminate superoxide anions ($O_2^-$) and organic radical cations (ORCs). More importantly, this phenomenon is generally found in different OSCs (Fig. 1d, EPR results of typical OSCs see Supplementary Section 3). These results suggest that sublimation only removes the physical adsorbed oxygen from OSCs, as observed in the measurements of XPS. However, the chemical adsorbed trace oxygen (i.e., $O_2^-$) is difficult to be removed. In addition, the first-principle molecular dynamics (FMD) simulations are performed, which suggest the trace oxygen is located in the molecular intralayer or interlayer of OSCs lattice (Supplementary Section 4), and hence these oxygen dopants are highly stable and de-doping of them becomes very difficult.

## De-doping trace oxygen from OSCs and devices

It is highly essential to de-dope these innate oxygen dopants to investigate their roles in OSCs and devices, and further optimize their optoelectronic properties for application. Here, an elaborately-controlled soft plasma treatment ($H_2$, $N_2$, or Ar) is developed to achieve this target (Supplementary Section 5). Surprisingly, under optimized treatment conditions (see Method), this de-doping process is nondestructive for OSCs as confirmed by their X-ray diffraction (XRD) and atomic force microscopy (AFM) results, etc. (Fig. 2a–c and Supplementary Section 6). Through this deoxygenation method, the de-doping effect is examined by quasi-in-situ ultraviolet photoelectron spectroscopy (UPS) and Time of Flight Secondary Ion Mass Spectrometry (TOF-SIMS). The UPS and TOF-SIMS are connected through a

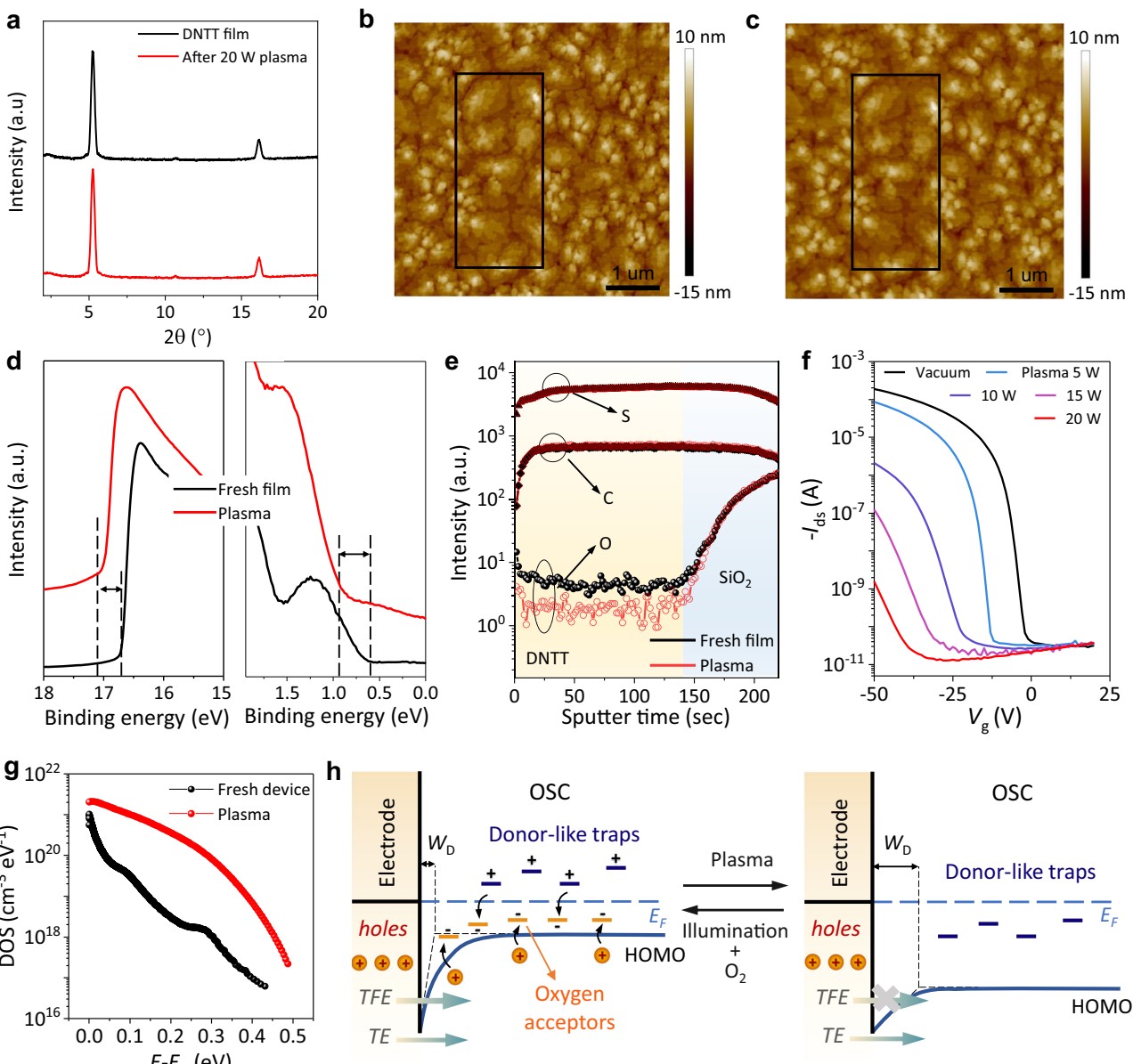

**Fig. 2 | De-doping oxygen dopants from OSCs and devices.** XRD spectra (**a**) and AFM images (**b**, **c**) of DNTT films before and after de-doping treatment, showing the non-destructive feature of the de-doping method. **d** UPS spectra at secondary electron cut-off region (left) and near the Fermi-level region (red) of the DNTT film before and after de-doping treatment, respectively. **e** Intensity of the TOF-SIMS data for the C, S, and O elements of the DNTT film before and after de-doping treatment, respectively. **f** Gradual removal of the *p*-type transfer characteristics of the DNTT OFET under de-doping treatment. $V_{ds} = -50$ V. $H_2$ plasma is used for these experiments. **g** Trap DOS of the OFET before and after plasma treatment extracted by FET method. **h** Energy band model of the OFET before and after de-doping. $W_D$ is the depletion layer width. TE and TFE denote thermionic emission and thermionic field emission, respectively. $\Delta E_B$ is the height lowering of Schottky barrier induced by image charges.

vacuum pipeline (see Methods), and all characterizations are performed in vacuum. For example, both the secondary electron cut-off and Fermi level of DNTT films shift about 0.35 eV towards high energy after the plasma treatment (Fig. 2d), suggesting the broadening of the gap between $E_F$ and HOMO, which is strong evidence to prove the success of the de-doping process. In addition, the density of state (DOS) and charge spatial distribution of DNTT films with and without oxygen incorporation evaluated by density functional theory (DFT) (Supplementary Section 4) show the similar tendency. In addition, TOF-SIMS results clearly show the decrease of oxygen concentration (Fig. 2e), while no changes are observed for other elements of C and S, further confirming the removal of oxygen dopant by this effective de-doping process.

Electrical properties of OSCs are significantly changed after the de-doping process, e.g., the pristine organic field-effect transistors (OFETs) show typical p-type characteristics with saturation mobility of 2 cm²·V⁻¹·s⁻¹ and $V_T$ of −5 V (Fig. 2f). After de-doping treatment, the transfer curves immediately shift negatively with a slight decrease in mobility and off-state current. The $V_T$ continue shifting negatively with increasing de-doping plasma power, suggesting a reduced carrier density in the conducting channel under the same gate voltage. Meanwhile, the subthreshold swing significantly deteriorates, implying the increase of the density of donor-like traps. To precisely extract the mobility in this process, we re-measured the transfer curves of the de-doped OFETs under higher gate voltage (Supplementary Section 7). Although the $V_T$ and hysteresis significantly increase, the decrease in mobility is not very severe (<50%), which indicates that mobility is a nature of materials and not significantly influenced by doping even considerable trap states exist inherently in OSCs. The distributions of DOS in the energy gap of OFETs before and after de-doping are quantitatively extracted by the classical FET method[30] (Supplementary Section 7). As shown in Fig. 2g, the DOS of the OFET becomes much higher from band edge to $E_F$ after de-doping, which implies that oxygen doping could pre-empty the donor-like traps in p-type OSCs. Then, the significantly increased contact resistance and the deteriorative bias stress also support this point (Supplementary Section 7). When p-type characteristics almost completely disappeared (the corresponding output curves of the de-doping process are shown in Supplementary Section 8), the conductivity is immeasurable (Supplementary Section 7). To further understand the role of oxygen doping in charge transport, the low-temperature measurements are performed. As a result, the temperature-dependent charge transport behavior of the fresh DNTT OFET accords well with the feature of the multiple trapping and release (MTR) model[31] while the de-doped device shows a typical hopping transport, which demonstrates that the increased trap states after de-doping dominates the charge transport (for detailed discussion, see Supplementary Section 7).

The energy band model of the OFET is outlined in Fig. 2h to elucidate these results. Oxygen acts as an acceptor in OSCs and its states distribute along the edge of the HOMO. Primarily, oxygen doping pre-empties the donor-like trap states[32] induced by static/dynamic disorder or other factors[33] (i.e., it releases the $E_F$ pinning) and thus yields mobile hole carriers in the HOMO (i.e., it renders the $E_F$ close to the edge of the HOMO), which is in analogy to the role of hydrogen in a-Si[34]. Meanwhile, the width of depletion layer and the height of the Schottky barrier at the metal/semiconductor interfaces is reduced due to the oxygen doping, which facilitates carrier injection through thermionic emission (TE) and thermionic field emission (TFE)[35]. After de-doping treatment, oxygen doping is eliminated, which shifts the $E_F$ away from the HOMO, broadens the depletion layer, and increases the barrier height. As a result, the concentration of hole carriers decreases sharply and charge injection is suppressed, causing the degradation in conductivity and contact resistance (Supplementary Section 7). Furthermore, the de-doping makes the pre-emptied donor-like trap states occupied, and the carrier transport is thus strongly impeded by

the trapping effect, causing the subthreshold swing deterioration. According to MTR model[31], a decrease in mobility is expected. At this point, even if a large gate voltage is applied, the $E_F$-HOMO energy difference remains large due to the $E_F$ pinning effect of the traps, and hence the threshold voltage dramatically increases. Remarkably, all OFETs based on different OSCs (Fig. 1d) show identical tendency after the de-doping treatments (Supplementary Section 9), demonstrating the general applicability of this de-doping process for OSCs and devices.

### Reversible doping/re-doping and de-doping OSCs and devices

As for the de-doped device shown in Fig. 2f, its p-type characteristics could be recovered slowly and partially when 1 atm of oxygen is filled in the vacuum system (Fig. 3a). However, when the OFETs are illuminated in oxygen atmosphere, its p-type characteristics can be recovered rapidly and almost fully (the output curves are shown in Supplementary Section 10), which suggests the indispensable role of light in re-doping process. The re-doping process could be practically inhibited if the de-doped OFET is encapsulated with the glass-glass technique (Supplementary Section 11), indicating that the de-doping state could be stable in suitable conditions. The switching curves in Fig. 3b, c show that the whole de-doping and re-doping processes can be cycled several times within 20 min when the ultrathin (approximately 5 nm) film is used (the influence of film thickness on the de-doping process are discussed in Supplementary Section 7), suggesting the excellent reversibility and high efficiency. The current of the OFET increases rapidly and exceeds the initial value after illuminated in $O_2$ (area 3 in Fig. 3b). The exceeding current is attributable to non-equilibrium photon-generated carriers. These extra carriers gradually decrease and disappear in a few minutes and the decline of the current gradually stops (area 4). In addition, different plasma treatments could achieve similar de-doping effect (Fig. 3b), but only $O_2$ exposure under illumination can realize fully re-doping effect (Fig. 3c), and the higher the energy of light, the faster the re-doping process (Fig. 3d, and Supplementary Section 11). The evolution of the EPR signals could more vividly depict the reversible de-doping and re-doping processes shown in Fig. 3e, f. The characteristic EPR signal of DNTT radical cations ($g = 2.0034$) and $O_2^-$ are detected in the fresh DNTT films (see Methods). Furthermore, the spin density $N_{spin}$ of ORCs (i.e., hole carrier concentration) is calculated approximately $1.9 \times 10^{15}$ cm⁻³ by the Curie law[36]. As expected, after plasma treatment, no EPR signal is observed, indicating the validation of our de-doping method. Furthermore, the EPR signals of ORCs and $O_2^-$ are almost completely recovered after this re-doping process. The reversible de-doping and re-doping could be depicted in Fig. 3g and Supplementary Section 12. Note that although the polymers are also investigated in this work and the similar de-doping and re-doping phenomena are found, the mechanisms behind these may be more complicated and inconsistent with that of small molecule system, due to the differences in their microscopic structures and energy band structures. The complexity in doping system makes it more challenging to study these processes in polymers.

### Modulating electronic properties of OSC devices by controllably doping or de-doping

To investigate the effect of oxygen doping/de-doping on n-type OSCs, PTCDI-$C_8$ is investigated firstly. Interestingly, when the PTCDI-$C_8$ OFET is treated by de-doping process, its electron transport is significantly enhanced, as shown in Fig. 4a, in which electron mobility is increased by over one order of magnitude to 2.2 cm² V⁻¹ s⁻¹, and the threshold voltage and $I_{on/off}$ ratio are also improved. Other 4 n-type OSCs (N2200, NDI-cy6, DHF-4T, and PDI-$CN_2$) show similar behaviors (Fig. 4b and Supplementary Section 13). These results suggest that oxygen dopants in OSCs severely inhibits electron transport, analogous to the impact of trace hydrogen on p-type behaviors in GaN[37], which can explain the general fact for OSCs that, comparing to their

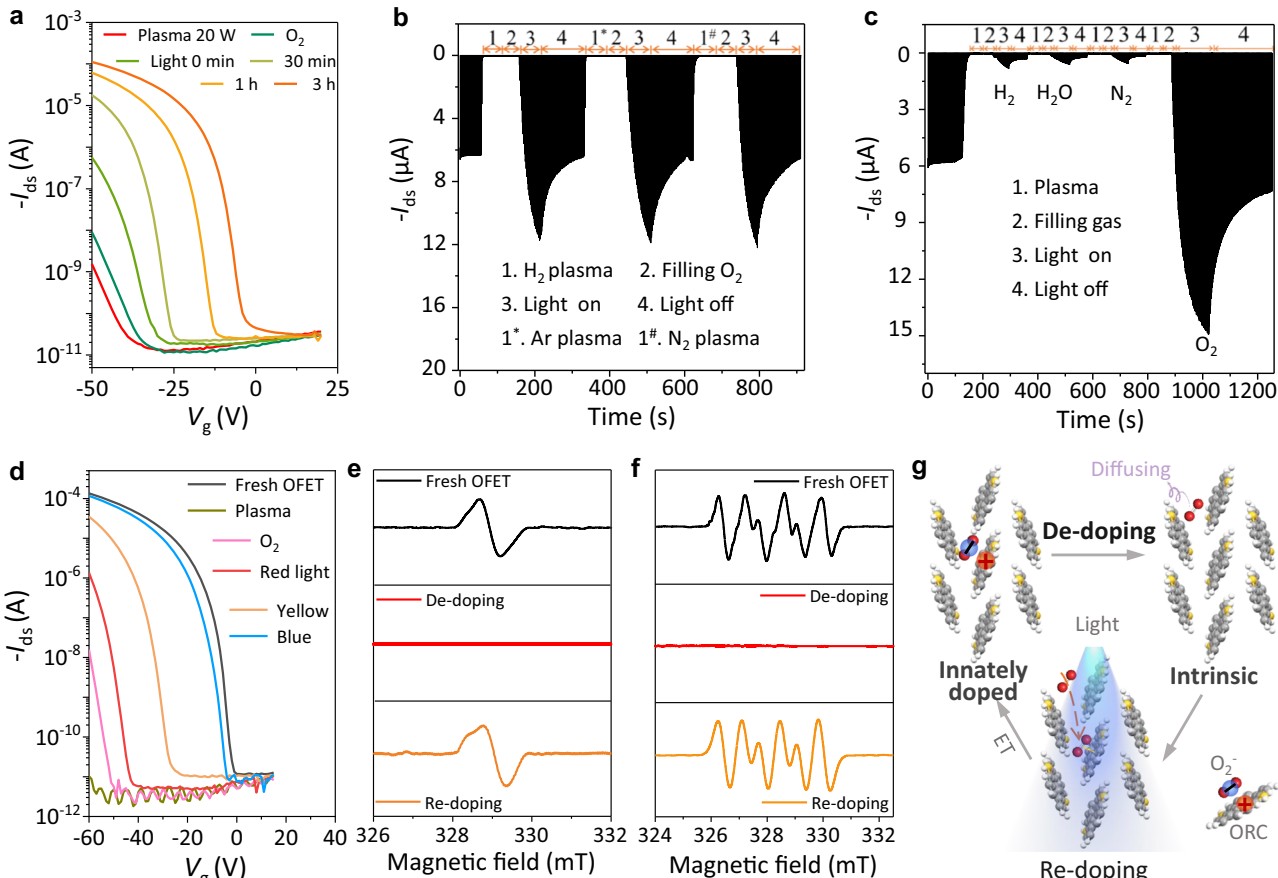

**Fig. 3 | Reversible doping/re-doping and de-doping OSCs and devices. a** Gradual recovery of the *p*-type transfer characteristics (i.e., re-doping process) of the de-doped DNTT OFET (Fig. 2f) under illumination in oxygen. $V_{ds} = -50$ V. The red curve of "Plasma 20 W" is the same as that in Fig. 2f. Switching cycle of the ultrathin (approximately 5 nm) DNTT OFET in multiple de-doping and re-doping processes with plasma treatment in H₂, Ar, and N₂ (**b**), and under illumination in H₂, H₂O (Ar), N₂, and O₂ (**c**). $V_{ds}$ is set as $-30$ V and $V_{gs}$ is switched from 0 to $-30$ V at 1 s intervals. The current shows a slight recovery when illuminated in H₂, H₂O (Ar is used as carrier gas through a gas-washing bottle with water to transfer H₂O), and N₂, which might be attributed to the residual O₂ in these gases (note that oxygen in present at ppm level in the purified gas (99.999%)). **d** Transfer curves of the ultrathin (about 5 nm) DNTT OFET in the re-doping process under the same duration of illumination with different wavelengths. Blue ~ 450–435 nm, yellow ~ 597–577 nm, red ~ 760–622 nm. $V_{ds} = -60$ V. The evolution of the EPR signals of ORCs (**e**) and O₂⁻ (**f**) in the process of plasma treatment and illumination in oxygen, showing the reversible elimination and recovery processes. **g** Schematic illustration of the de-doping and re-doping processes, showing the switch between innately doped state and de-doped state by eliminating (plasma treatment) and regenerating (illuminated in O₂) ORCs and O₂⁻ in OSCs. ET denotes electron transfer.

*p*-type partners, *n*-type candidates always show poor performance, even in vacuum or inert atmosphere. The mobility and threshold voltage of the organic semiconductors used in this work in the doped and de-doped states are extracted with the large overdrive voltages in Supplementary Section 14.

Moreover, the feasibility is opened for OSCs and their devices based on the reversible doping/re-doping and de-doping methods, e.g., modulating their key electronic properties in a nondestructive way. To demonstrate this potential, three kinds of modulations are performed. Firstly, as a vital parameter of OFETs, $V_T$ is difficult to be controlled precisely. Based on this strategy, as shown in Fig. 4c, d, all OFETs no matter with negative $V_T$ or positive $V_T$ could be modulated to turn on at approximately 0 V, without the observed decrease in mobility. Secondly, it is crucial to control the doping concentration to tune the properties of organic devices. Attractively, efficient doping is achieved with almost no decrease of mobility by this doping or re-doping process, e.g., after re-doping the conductivity ($\sigma$) of C₁₀-DNTT could be sharply increased over two orders of magnitude up to 72 S m⁻¹, and the mobility only shows a slight decrease (5.2 to 3.6 cm²·V⁻¹·s⁻¹, Supplementary Section 15). With these data, the carrier density *n* is calculated as $2 \times 10^{18}$ cm⁻³ (using $\sigma = ne\mu$), representing high doping level for small molecular OSCs. Thirdly, the

polar type of the transport behaviors can be modulated by this de-doping and re-doping strategy. For example, OFETs of TIPS-pentacene show *p*-type behavior both in air and vacuum[38], but they can be modulated to be *n*-type via our de-doping process (Fig. 4e, f), and certainly can be recover to *p*-type again by our re-doping process. In addition, it is well known that N2200 is a high-performance *n*-type OSC[39], which can be endowed with *p*-type behavior via our re-doping strategy (Fig. 4g, h), and certainly can be covered to *n*-type again by de-doping process. The inversion is achieved by compensating the electrons in *n*-type OSCs by excess oxygen doping. Note that these polarity conversions are achieved without special optimization processes, such as energy-level alignment[40] and interface trap passivation[14]. Although oxygen doping in OSCs caused by air exposure has been found and discussed over 20 years, no one noticed that trace oxygen is inherent in purified OSCs even without any exposure. As a result, many phenomena, such as the $E_F$ closer to the HOMO[41,42], carrier densities and conductivities beyond expectation[43–45], the deterioration of subthreshold swing (*S*) at low temperature[46,47], unconfirmed effect of ionization gauges on charge transport[48,49] (see Supplementary Section 16) have been perplexing. Now, our findings unravel and overcome those long-standing confusions and challenges in OSCs and organic electronics.

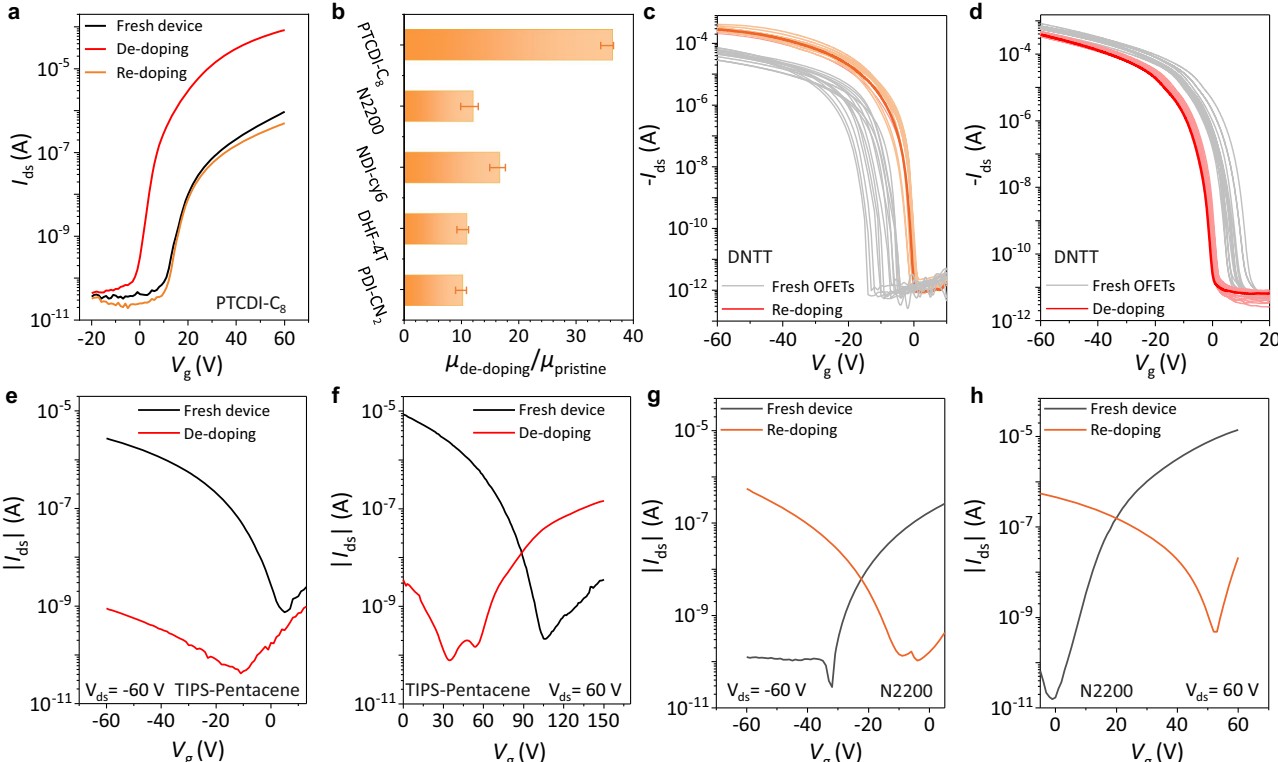

**Fig. 4 | Modulating electronic properties of OSC devices by doping or de-doping. a** Transfer curves of the PTCDI-$C_8$ OFET, showing that removal of oxygen by de-doping significantly enhances electron transport and re-doping reverses the process. **b** Mobility increases of 5 typical $n$-type OSCs after de-doping. The statistics of error bars are a count of 5 devices per molecule. **c, d** Precise modulation of the threshold voltage of the $C_{10}$-DNTT OFETs (from negative (**c**) and positive (**d**) values to near zero). **e, f** Conversion of TIPS-pentacene OFET from $p$-type to $n$-type conduction by de-doping. **g, h** Conversion of N2200 OFET from $n$-type to $p$-type conduction by re-doping.

Our study develops an uncommon soft plasma deoxygenation method to remove trace oxygen in OSCs, and reveals that trace oxygen, as inherent dopants in OSC lattice, significantly pre-empties the donor-like traps in OSCs, and is the origin of the $p$-type characteristics of OSCs and the reason why $n$-type property is significantly inhibited. This finding explains previously perplexing organic electronics phenomena and will enable a foundation upon which we can re-understand charge transport in OSCs. Furthermore, the reversible de-doping and re-doping methods are facile, green and efficient, making the dream of precisely modulating optoelectronic properties of OSCs and devices approachable. We believe these methods will become a core technology for the growing organic electronics industry.

## Methods
### Materials
OSCs are purchased from Sigma-Aldrich, Shanghai Daeyeon Chemicals, and TCI Chemicals. All small molecules are purified by physical vapor transport three times. The semiconductors in different batches and from different companies exhibit the same phenomena. Polystyrene (PS), polymethyl methacrylate (PMMA) and octadecyltrichlorosilane (OTS) are purchased from Sigma-Aldrich. High purity metals (99.95%) are from Beijing XXBR Technology Co., Ltd.

### Home-made system
For in-situ measurements during the de-doping and re-doping processes, the system integrates a DC glow plasma source (Supplementary Section 5), an optical window, and the environmental control system, and then is connected to a $N_2$ glovebox. The environmental control system controls the type and velocity of the filling gases and modulates the atmospheric pressure in the range of $10^{-5}$–$10^5$ Pa. All gases used in this work (including $O_2$ and $H_2O$) enter the home-made system through a soft polyurethane (PU) tube. To avoid the cross contamination in the PU tube, the purified $H_2$, $N_2$ and Ar gases (99.999%) for de-doping treatment are further purified by the dehydration tube and deoxygenation tube in series. The DC glow plasma source normally works on low power (below 20 W), leading to low ionization ratio and low electron density (i.e., soft plasma). All the equipment is off-the-shelf. For technical details, see Supplementary Section 5.

### Fabrication of OFETs
OFETs in this work adopt a bottom-gate top-contact configuration, unless otherwise stated in the text. The highly doped Si wafers (500 μm thick) with a 300-nm-thick thermal oxide layer are treated by oxygen plasma (100 W, 30 s) to produce hydroxyl groups, and then modified with OTS in vacuum oven for 1 h at 120 °C. DNTT, $C_{10}$-DNTT, PEN, DPA, and CuPc are vapor deposited on the OTS-treated $SiO_2$/Si surface at a ratio of 0.1 Å $s^{-1}$ below $10^{-4}$ Pa (achieving a usual thickness of 20−30 nm), and substrate temperature is kept between 60−100 °C. PTCDI-$C_8$, NDI-cy6, DHF-4T, and PDI-$CN_2$ are vapor deposited on the PS-coated $SiO_2$/Si (5 mg $ml^{-1}$ in toluene, spin-coated on $SiO_2$ at 3000 rpm for 1 min by a KW-4A, Setcas) surface at a ratio of 0.1 Å $s^{-1}$ with the thickness of 20−30 nm, and substrate temperature is kept between 60 °C. The film of Tips-pentacene is prepared by the drop-casting method (1 mg $ml^{-1}$ in toluene) onto the PS-coated $SiO_2$/Si. P3HT and N2200 (P(NDI2OD-T2)) are spin-coated on the OTS-treated $SiO_2$/Si (10 mg $ml^{-1}$ in chlorobenzene, 2000 rpm for 1 min), and then annealed at 100 °C in $N_2$ for 2 h. A 20 nm layer of Au is deposited as the source-drain electrodes through a shadow mask under a vapor deposition ratio of about 0.005 nm $s^{-1}$. The ultrathin DNTT film (about 5 nm) is vapor deposited on the PS-coated $SiO_2$/Si at a ratio of 0.02 Å $s^{-1}$, and the substrate is kept at room temperature. Note that the uniformity

and stability of the ultrathin films are usually poor, so regular thickness films are used for most of this work. All deposition processes of OSCs are performed in the vacuum thermal evaporation system connected to a $N_2$ glovebox, convenient for the storage of devices in an oxygen-free environment. To fabricate the bottom-gate bottom-contact OFETs, 20 nm Ag electrodes are deposited on OTS-treated $SiO_2$/Si, and then modified with pentafluorobenzenethiol (PFBT, purchased from Aldrich) by immersing into a 10 mM solution of PFBT in 2-propanol for 5 min. In addition, rubrene single crystals are grown by physical vapor transport, and then transferred onto a prefabricated PMMA/Si stamp with an air gap. PMMA (10 mg ml$^{-1}$ in toluene) is spin-coated on the highly doped Si at 800 rpm for 1 min, and then 20 nm Au is deposited on PMMA/Si. The air gap is fabricated by mechanical scraping using a probe. In addition, the $C_8$-BTBT crystalline film is prepared by the space-confined self-assembly method, and the Au (80 nm) stripes are stamped on the $C_8$-BTBT crystalline film as the source and drain electrodes. The channel width/length for the thin-film transistors and conductivity measurements in this work is 5:1. For the cases of crystal transistors, e.g., Rubrene and $C_8$-BTBT, the W/L is 2:1.

### In-situ electrical characterizations

The electrical characterizations of the OFETs are carried out in the home-made system using two types of semiconductor device analyzers (an Agilent B1500A and a PDA FS380). The leading wires are bonded on the source-drain electrodes and the gate electrode by a wire bonder. The gases of the plasma treatment for de-doping are $H_2$ or $N_2$ or Ar, and for re-doping is $O_2$. The time of the plasma treatment for de-doping is usually below 1 min. Illumination is performed using a Xe lamp (adjustable power of 0−300 W) and an LED Light (15 W) with switchable wavelength (450 ~ 435 nm, 597 ~ 577, 760 ~ 622 nm). In the de-doping and re-doping investigations, the OFETs are measured about 1 min after the plasma treatment and examined about 3 min after illumination in $O_2$ (to avoid the influence of photogenerated carriers on the results).

**Reversible de-doping and re-doping test**. To accelerate the multiple de-doping and re-doping processes, ultrathin (about 5 nm) OSC films are used, with which the de-doping and re-doping process can be repeated several times in 20 min. The switching cycle of the ultrathin DNTT OFET in the multiple de-doping and re-doping processes is measured by a PDA FS380. $V_{ds}$ is set to −30 V and $V_{gs}$ is switched from 0 to −30 V with intervals of 1 s. The power of the plasma is set to 1 W.

### EPR characterizations

X-band (9.5 GHz) electron paramagnetic resonance (EPR) measurements are performed with both a Bruker EMXplus-6/1 and JEOL JES-FA200. The microwave power and the modulation magnetic field are carefully adjusted to produce the optimal EPR signal. The raw material is purified by vacuum sublimation, and then stored in a dark $N_2$ glovebox. The purified powder is loaded into a glass capillary in a $N_2$ glovebox. After sealing with silicone grease, the glass capillary is inserted into a quartz tube for EPR testing. To detect $O_2^-$, the purified powder is dispersed into DMPO diluent (volume ratio of 1:100 in methanol or dimethylsulphoxide) in a $N_2$ glovebox for at least 30 min. Then the suspension is sucked up with a glass capillary. After sealing with silicone grease, the glass capillary is inserted into a quartz tube for EPR testing. For examine films, the fresh DNTT films are deposited onto a 3.5 mm wide × 20 mm long quartz substrate and stored in a $N_2$ glovebox. The samples are then inserted into a quartz EPR tube. The signal of the DNTT film is usually weak due to the larger spin-orbit coupling caused by the sulfur atoms. The signal intensity can be improved by increasing the quantity of samples. The detection of $O_2^-$ of the films is similar as the above. For low-temperature measurements, a continuous liquid nitrogen flow cryostat is used to control temperature. 2,2-Diphenyl-1-picrylhydrazyl (DPPH) is used as a standard spin counting reference.

To track unpaired electrons in the de-doping process, the plasma-treated DNTT film (not exposed to the air) is inserted into a quartz EPR tube in the $N_2$ glovebox and then the tube is sealed with silicone grease. The samples for $O_2^-$ detection of the plasma-treated DNTT films are prepared and sealed in the glovebox as described above. To investigate the re-doping process, the plasma-treated DNTT films are illuminated in $O_2$ with a Xe lamp for 24 h, and then the tests are performed as above. Some weak signals are carefully denoised.

### Quasi-in-situ UPS and TOF-SIMS characterizations

UPS (PHI 5000 Versaprobe II) and TOF-SIMS (TOF.SIMS5-100) are connected through a vacuum pipeline, whose environmental chamber (which controls the gas type and the atmospheric pressure) is integrated with a plasma source and connected to a glovebox that is used to transfer samples without air exposure. DNTT films (about 50 nm) are deposited onto the highly doped Si and $SiO_2$/Si, and sealed into a glass bottle in a $N_2$ glovebox for UPS and TOF-SIMS tests. The excitation source for UPS is He Iα ($h\nu$ = 21.22 eV). Vacuum level shifts are determined from the low kinetic energy part of UPS spectra with a −5 V sample bias. Secondary ions employed $Cs^+$ as the primary ion source. The size of the crater is 100 × 100 um, and the area of acceptance is 25% of the total sputtered area. The UPS and TOF-SIMS tests are performed on the fresh DNTT films in turn, and they are returned to the environmental chamber through a vacuum pipeline to be treated by plasma with purified gases. The plasma-treated DNTT films are transferred to the UPS and TOF-SIMS in turn through a vacuum pipeline for testing. All the transfer processes of samples are carried out in vacuum. The in-situ test equipment and technology supports are provided by Vacuum Interconnected Nanotech Workstation (NANO-X) in Suzhou.

### Morphology, physical phase, and chemical structure characterizations

Morphology, physical phase, and chemical structure characterizations on the DNTT films are performed before and after plasma treatment. AFM measurements are carried out on a Dimension ICON (Bruker). XRD measurements are carried out on a MiniFlex600 (Rigaku). DNTT films are deposited on a quartz plate for the measurement of the UV-vis absorption spectrum (a Lambda 750) and deposited on Au-coated Si for the measurements of the Raman spectrum (a DXR Microscope with a 532 nm laser), and IR spectrum (Nicolet IN10, Thermo Fisher Scientific). For the UV-Vis test, the samples are loaded into a quartz colorimetric ware and sealed by silicone grease. A 50 nm DNTT film is deposited on highly doped Si in a vacuum thermal evaporation system and transferred under the protection of $N_2$ to a $N_2$ glovebox connected to an X-ray photoelectron spectrometer (ESCALAB-250Xi, Thermo Fisher Scientific).

### Modulation of the key device parameters

**Threshold voltage.** In-situ electrical measurements are performed on 14 DNTT (20−30 nm) OFETs with positive $V_T$ after plasma treatment. The parameters of the plasma treatment are carefully modulated until the $V_T$ is tuned at about 0 V. Correspondingly, 14 DNTT OFETs with negative $V_T$ are repeatedly measured after simultaneous annealing and illumination in oxygen until the $V_T$ is tuned at about 0 V.

**Conductivity.** The $C_{10}$-DNTT OFET is annealed at 80 °C in a cavity with an inspection window, and $O_2$ is filled into the cavity with a pressure of about 3 atm. Illumination is performed by a Xe lamp.

**Polarity.** The Tips-pentacene OFET is treated by plasma at 20 W for 10 min and P(NDI2OD-T2) OFET is annealed at 60 °C and illuminated in $O_2$ for 3 h.

**Electron mobility.** The PTCDI-$C_8$, N2200, NDI-cy6, DHF-4T, and PDI-$CN_2$ OFET are treated by plasma at 5 W for 1 min.

## Data availability

All data supporting the findings of the study, including experimental procedures and characterizations, are available within the paper and its Supplementary Information, or from the corresponding author upon request.

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

## Acknowledgements

The authors are grateful to National Key Research and Development Program (2018YFA0703200), National Natural Science Foundation of China (52225304, 52073210, 52203236, 52121002), Natural Science Foundation of Tianjin City (19JCZDJC37400, 19JCJQJC62600). The authors sincerely appreciate the technological support from Vacuum Interconnected Nanotech Workstation (NANO-X), Suzhou Institute of Nano-Tech and Nano-Bionics (SINANO), Chinese Academy of Sciences. The authors are grateful to Haihe Laboratory of Sustainable Chemical Transformations for financial support.

## Author contributions

L.L. and Y.N.H. discovered the oxygen de-doping and re-doping of organic semiconductors and conceived the experiments. K.W. found the acceleration effect of illumination on re-doping. Y.N.H. and X.C. performed the device fabrication, electrical and spectral characterizations, and developed the controlled doping and de-doping technologies. Y.S. conducted the theoretical calculations. Y.X.H., Z.W., L.Y., S.W., D.J., X.Z., H.D., X.W., Z.L., R.H., Z.G. and Y.C. assisted in the experiments. L.L., Y.N.H., X.C. and W.H. conceived, analyzed, and wrote the manuscript. L.L. and W.H. supervised the work.

## Competing interests

The authors declare no competing interests.
