## [Peer Review File · Nature Communications]

REVIEWER COMMENTS

Reviewer #1 (Remarks to the Author):

In the present study, the author proposes a new deoxygenation process in order to dope/de-dope/ and re-dope organic semiconductor materials in a very reliable way. The study is very interesting with results that are extremely interesting for other researchers in the field but also the emerging industry for printed electronics. The results from this very systematic and detailed work have the potential to influence the entire field of organic semiconductors. For example, many of the transport models and material parameters (e.g., trap distribution, intrinsic mobility) might need to be revised based on the outcome of this study. Hence, I enthusiastically support this study for publication in Nature Communications. However, I would like the authors to elaborate on the soft-plasma process in order to understand how generalizable their findings are. Furthermore, a few key parameters of the process need to be disclosed in order to allow other researchers to scrutinize the findings of this paper.

More detailed criticism is listed below (not sorted by major/ minor):

1) „However, there is no doubt that trace oxygen residues are unavoidable in these processes.“ After reading the paper I agree with the statement of the authors but can you please elaborate on why methods such as vacuum sublimation fail/ or oxygen residues are unavoidable? Furthermore, looking at the TOF-SIMS results, I am concluding that the O₂ residual concentration has been reduced by one order of magnitude to $\sim 10^{14} \text{cm}^{-3}$. Is that correct? Is that the lower limit of detection of the TOF-SIMS or is there still oxygen left in the OSC? How to remove these residuals? Furthermore, why do the C and S signals do not disappear once you get to the SiO₂ layer.

2) Please elaborate on the plasma process. Why does it happen? Why you do not see a difference for H, Ar, N? What exactly is the role of the light during the recovery? Wouldn't blue light and O₂ trigger an irreversible photooxidation? What are the details of the plasma process, e.g., you mention a wide pressure range of the system in the paper, but which pressure did you use for the treatment - it is not really specified in the paper? What is the voltage you have at the anode/ cathode during the plasma? At which power density do you start seeing etching/ damages at the OSC due to ion bombardment etc., e.g., what happens at 30,40...200W?? Is the plasma uniform in the home-built reactor/ did you please the samples directly in the plasma? Did you also test a low-power RF plasma process?

3) Fig. 2 → What is the gas used for these experiments? Mention it in the caption, otherwise, the reader has to search for this information first

4) How do you know that the donor-like traps exist in the OSC beforehand and are not created by the plasma? As you are showing the effect of the "plasma-annealing" for several materials, would the authors agree with the generalization that these donor-like traps exist in the majority of OSCs? What could be their origin?

5) "Purified H₂, N₂, and Ar gases (99.999%) for de-doping treatment and further purified ..." Is this additional purification needed? Any indication for that?

- 6) I would appreciate it if the authors could add a summarizing table to the SI including all mobility values & threshold voltages for all materials in the doped and undoped states.
- 7) In order to understand the impact of the plasma treatment on industrial applications, it would be very valuable to know how long it takes for the redoping process to disappear under state-of-the-art glass-glass encapsulation conditions. For example, de-doped devices might show a much stronger bias stress due to rapid redoping. If this is the case, then the "plasma-annealing" might not be a desirable process as doped devices will not undergo this transition.
- 9) Geometry details for the transistors & conductivity measurements should be reported (W/L etc.)
- 10) The mobility extracting for the de-doped hole transport FETs is in my opinion not valid as you are not comparing the devices at the same overdrive voltage. For example, a good measurement of the mobility for de-doped devices should be done e.g., at -100V V_{GS}
- 11) Fig 3b: 'switched from 0 to -30 V at 1 s intervals' - why do you see this continuous decrease of current with time. Is that bias-stress
- 12) Do the authors have any confirmation that sublimation & vacuum deposition does not give O₂-free films. From my experiments I can say that a DNTT film in UHV is not conductive but it becomes conductive in a glovebox. Hence, sublimation is a suited deoxygenation technique to clean materials, however, it cannot be applied to films that have been already fabricated  that is the advantage of the plasma
- 13). Fig. S12: Plasma de-doping 'with the energy barrier to charge transfer overcome by the plasma'. Why is this not possible by heat or light, but only plasma? What is the reason for using H₂? Shouldn't inert gases be favored?
- 14) In my opinion, the reason for the unique characteristic of removing O₂ trace amounts by plasma is not well described. The main text completely lacks any such description and in the SI, it is only briefly mentioned. Why do other methods like thermal annealing/annealing under inert atmospheres/illumination of OSCs fail to accomplish the same (were they also tested)?

Reviewer #2 (Remarks to the Author):

The authors report on oxygen trace effects on organic semiconductors in terms of charge transport. The results are interesting and the implication would be versatile. However, a few issues should be addressed before the potential publication in this journal.

Lack of stability study:

The authors mentioned at the beginning that doping is important to stability, and indeed, doping/de-doping can be observed in the variation of device stability. It is necessary to examine stability and to include the stability results.

Difference in microscopic structure:

It should be noted that the interaction between oxygen and organic semiconductor might be quite different for polymers and small molecules. It means, C8-BTBT and P3HT, for example, are rather different from a microscopic point of view. It's unlikely that the same rule applies to both. So, the differences should be identified to figure out why the corresponding doping/de-doping arises.

Impact of light:

The authors hint at light's effect on de-doping and re-doping, but without experimental result. It is necessary to check the role of light playing in those processes and to see if it has any consequences for the stability of the doped/undoped states. Perhaps do some stability tests to clarify the underlying mechanism by avoiding the influence of photo-generated carriers.

Need for low-temperature measurements:

To gain better understanding of the oxygen's role in the context of charge transport, it is necessary to perform low-temperature measurement. Such data could provide additional insight into the interaction between oxygen and charge carriers in organic semiconductors.

Consideration of film thickness:

In single crystals (including C8-BTBT and DNNT, often called crystalline systems), oxygen may be just absorbed on the first layer. Thus, it is necessary to investigate the impact of film thickness in considering factors such as channel exposure and device structure. The manuscript should include data or discussion regarding the effect of varying thickness on oxygen-related phenomena, especially for small molecules.

Need for presentation improvement:

The English is fine, but improvement is still needed. Some mistakes or improper statements should be revised. 'Underlaying' should be 'underlying'. 'Saturated mobility' is better to be 'saturation mobility' or 'mobility in the saturation regime'. 'In the forbidden band of OFETs' is not correct. OFETs have not an exact forbidden band. 'on/off rate' is often called 'on/off ratio'. The statement of 'the reversible de-doping and re-doping methods are clean' is not good.

Furthermore, usage of adjective and adverb should be conservative. There are too many in a sentence, e.g., the 2nd paragraph, page 11.

In summary, the manuscript presents interesting findings regarding the role of trace oxygen in organic semiconductors, but it requires further refinement to address the aforementioned concerns and provide a more comprehensive understanding.

Reviewer #3 (Remarks to the Author):

In this manuscript the authors discuss the impact of trace levels of oxygen on the properties of organic field-effect transistors. They propose an efficient method for de-doping which they then use in conjunction with various p-type semiconductors to flip the conduction to n-type. The doping is reversible and the p-type conductivity is regained upon re-doping. Given the fact that even very small quantities of oxygen make a huge difference, it is most likely that many, if not most of the past measurements have been conducted on organic semiconductors that have been unintentionally doped with oxygen. This work provides new insights on the impact of oxygen, a ubiquitous dopant, and hence is relevant not only for transistor research, but all devices based on organic semiconductors, and especially those operating in air. I recommend publication after the following concerns have been addressed:

1. The de-doping process is critical to this work, but the authors provide vague details about it. It is very important that this method is described in more detail so that other groups are able to reproduce it.
2. There is quite a bit of prior work that discussed oxygen doping in organic semiconductors and that has not been mentioned. The intro can be misleading for a non-expert in organic electronics, who might draw the inaccurate conclusion that this is the first discovery of oxygen doping in organic semiconductors.
3. The discussion around Figure 4 is key to the entire work, but it is very short. More details about these measurements should be provided. I understand how a p-type semiconductor can become n-type, but how does an n-type become p-type upon de-doping?
4. Can the authors be more quantitative on the oxygen quantity that would be needed to obtain p-type transport? Is that quantity dependent on the material? How?
5. How does the contribution of doping on the channel properties compare with that on the contact resistance?

Reply for Reviewer #1

Your general comments: In the present study, the author proposes a new deoxygenation process in order to dope/de-dope/ and re-dope organic semiconductor materials in a very reliable way. The study is very interesting with results that are extremely interesting for other researchers in the field but also the emerging industry for printed electronics. The results from this very systematic and detailed work have the potential to influence the entire field of organic semiconductors. For example, many of the transport models and material parameters (e.g., trap distribution, intrinsic mobility) might need to be revised based on the outcome of this study. Hence, I enthusiastically support this study for publication in Nature Communications. However, I would like the authors to elaborate on the soft-plasma process in order to understand how generalizable their findings are. Furthermore, a few key parameters of the process need to be disclosed in order to allow other researchers to scrutinize the findings of this paper.

More detailed criticism is listed below (not sorted by major/ minor):

Our reply: Thank you very much for your high comments and glowing recommendation on our work! We also appreciate your professional and constructive criticism on the unclear points in this manuscript, which really help us to improve it!

Below are the point-to-point replies.

Your comments 1: “However, there is no doubt that trace oxygen residues are unavoidable in these processes.” After reading the paper I agree with the statement of the authors but can you please elaborate on why methods such as vacuum sublimation fail/ or oxygen residues are unavoidable? Furthermore, looking at the TOF-SIMS results, I am concluding that the O₂ residual concentration has been reduced by one order of magnitude to $\sim 10^{14} \text{cm}^{-3}$. Is that correct? Is that the lower limit of detection of the TOF-SIMS or is there still oxygen left in the OSC? How to remove these residuals? Furthermore, why do the C and S signals do not disappear once you get to the SiO₂ layer.

Our reply: Thanks for your professional comments! We appreciate that you agree with our viewpoint. We respond to your questions in the following three points.

(1) Why methods such as vacuum sublimation fail/ or oxygen residues are unavoidable? Given the low dielectric constant of OSCs (in the range of 3 to 4), the Coulomb interactions between ionized hosts and dopants (organic radical cations and O₂[•] in this work) are on the order of several 100 meV (*Acc. Chem. Res.* **2016**, 49, 370–378, **Fig. R2**), which implies that this interaction may be not broken even after sublimation (organic small molecules are generally sublimated < 400 °C, *i.e.*, < 60 meV). Accordingly, sublimation is not an efficient de-doping method, usually resulting in trace residual impurities or dopants OSCs (*Adv. Mater.* **2017**, 29, 1703063, **Fig. R2**). This point is supported by our experimental data and literatures.

Experiments: XPS measurements suggest that sublimation significantly reduces the oxygen concentration in OCSs films (**Fig. R1a** and **b**). However, the results of TOF-SIMS show that oxygen concentration in the sublimated film (fresh vacuum deposited DNTT film, not exposed in air) is further reduced by plasma treatment (**Fig. R1c**). Furthermore, EPR results also support this. The geminate organic radical cations (**Fig. R1d**) and $O_2^{\cdot-}$ (**Fig. R1e**) exist in the fresh DNTT films and are removed after plasma treatment. Therefore, we believe that the sublimation or vacuum annealing only remove the physical adsorbed oxygen, which is observed in the measurements of XPS. However, the chemical adsorbed trace oxygen (*i.e.*, $O_2^{\cdot-}$) is difficult to be removed.

Figure R1. XPS, TOF-SIMS and EPR measurements of DNTT films. The fresh DNTT film is sealed in a glass bottle in a N_2 glovebox and transferred into XPS and TOF-SIMS that connect to a N_2 glovebox. Inserts in **a** and **b** are the fine spectra of oxygen (around 532 eV). The geminate organic radical cations (**d**) and superoxide anion, *i.e.*, $O_2^{\cdot-}$ (**e**) exist in fresh DNTT films and are removed after plasma treatment.

Literatures: Jacobs and Moulé discussed thermal de-doping in their review article about molecular doping (*Adv. Mater.* **2017**, *29*, 1703063), and they pointed that thermal de-doping is commonly observed in films doped by F₄TCNQ, but **does result in incomplete remove of F₄TCNQ (Fig. R2)**.

a Given that the low dielectric constant of organic semiconductors (in the range of 3 to 4) results in considerable electron–hole pair binding energies on the order of several 100 meV,³⁴ it is important to know where the IPA levels are in

b Thermal dedoping is commonly observed in films doped by F4TCNQ,^[34,87,273] but does not result in complete recovery of film fluorescence.^[87]

Figure R2. (a) Discussion of Coulomb interaction in OSCs. (*Acc. Chem. Res.* **2016**, 49, 370–378). (b) Discussion of thermal de-doping by Jacobs and Moulé. Fluorescence is very sensitive to doping and its incomplete recovery indicates residual dopants (*Adv. Mater.* **2017**, 29, 1703063).

Eyer *et al.*'s work well supports our point (*J. Phys. Chem. C* **2017**, 121, 24929–24935); they found the EPR signals in a lot of p-type OSCs, caused by oxygen doping, are not reduced after vacuum sublimation (Fig. R3). Furthermore, Daniel *et al.* investigated oxygen incorporation in rubrene single crystals by an isotope (O^{18}) labelling method, and demonstrate that **without a significant exposure to light, O^{18} cannot incorporate in rubrene for periods of greater than a year** (*Sci. Rep.* **2014**, 4, 4753). However, they unexpectedly found that **large amounts of O^{16} are present in rubrene single crystal whether or not it is exposed to light.**

Figure R3. a, Spin-counting data for all 14 molecules (and postsublimation data for eight). b, Chemical

structures of the molecules. Concentration of atomic oxygen into the single crystal rubrene with (c) and without (d) light exposure.

- (2) **The results of TOF-SIMS in this work can only be used for qualitative analysis, i.e.,** TOF-SIMS measurements show the decrease of oxygen concentration but cannot determine the exact change in value because the signal intensity of oxygen is not necessarily proportional to oxygen concentration. *By the way, to make quantitative analysis with TOF-SIMS, several OSCs films with different known oxygen content must be prepared first as standard samples. It is very difficult with our current technology.*

In fact, the oxygen concentration in naturally doped OSCs is determined by the commonly-used electron paramagnetic resonance spectroscopy (EPR) method. Assuming all the doped oxygen is ionized, then the oxygen concentration is equal to $O_2^{\bullet-}$ concentration. And a $O_2^{\bullet-}$ production must be accompanied by an organic radical cation, so we approximate the concentration of organic radical cations to oxygen concentration. According to the Curie law, the peak area of the EPR signal is proportional to radical concentration,

$$\chi_{\text{Curie}} = \left(\frac{\mu_B^2}{3kT} \right) N_{\text{spin}} g^2 S(S + 1) \quad (1)$$

where k , N_{spin} , g , and S are the Boltzmann constant, total radical concentration, g -factor and the spin quantum number ($S = 1/2$). Therefore, $O_2^{\bullet-}$ concentration in OSCs can be determined by comparing the peak area of the EPR signal with the standard sample. Regrettably, the EPR signals of organic radical cations and $O_2^{\bullet-}$ cannot be detected in the de-doped OSCs, indicating their **concentration is below the detection limit**. Although oxygen signal in TOF-SIMS is still observable, the noise is already noticeable, which means the oxygen concentration is approaching the detection limit. Hence, we cannot determine the exact oxygen concentration in OSCs after de-doping but we can affirmatively say this concentration is extremely low.

- (3) **Why do the C and S signals do not disappear once you get to the SiO₂ layer?** It is because the OSC films used for TOF-SIMS are polycrystalline. The thickness at grain boundaries is thinner than that of the grains. When the molecules at grain boundaries are totally exfoliated by ion beam, the signals of SiO₂ layer appear, but the molecules of grain are still coated on the substrate. Therefore, the C and S signals can together appear with SiO₂ signals.

Our revision: The corresponding discussion of oxygen removal and thermal de-doping are added into Page 4 of main text and Supplementary Section 1.

Your comments 2: Please elaborate on the plasma process. Why does it happen? Why you do not see a difference for H, Ar, N? What exactly is the role of the light during the recovery? Wouldn't blue light and O₂ trigger an irreversible photooxidation? What are the details of the plasma process, e.g., you mention a wide

pressure range of the system in the paper, but which pressure did you use for the treatment - it is not really specified in the paper? What is the voltage you have at the anode/ cathode during the plasma? At which power density do you start seeing etching/ damages at the OSC due to ion bombardment etc., e.g., what happens at 30,40...200W?? Is the plasma uniform in the home-built reactor/ did you please the samples directly in the plasma? Did you also test a low-power RF plasma process?

Our reply: Thanks for your professional and meticulous questions! We will address your comments in two parts as follow:

(1) The details of the plasma process in de-doping. For clarity, we would like to address these questions with a technical sheet (**Table 1**).

	DC	RF	Thermal
Plasma	Low density and energy	High density and energy	Low efficiency,
Pressure Range	10-1000 Pa	10-100 Pa	< 1 Pa
Power	0-300 W	0-100 W	0-20 W
Atmosphere	H ₂ , Ar, N ₂ (H ₂ optimum efficiency, No difference for Ar and N ₂)		
Treatment Results	Ineffective (< 5 W, > 1000 Pa) Nondestructive (< 50 W, > 50 Pa) Destructive (> 50 W, < 50 Pa)	Destructive (all conditions)	Ineffective PS: It can be only used for ultrathin films

Firstly, we discuss the plasma source tested in this work, *i.e.*, DC plasma, RF plasma and thermal plasma (*e.g.*, we used a heated tungsten wire to ionize the thin gas in the vacuum chamber). DC source yields plasma with low density and energy, which is applicable for organic materials. RF source yields plasma with high density and energy. We had tried to use RF source at the power of 1 W to treat OFETs, but the electrical performance degraded and cannot be recovered. Thermal source is so inefficient that it only causes a slight shift of subthreshold voltage This effect can be significant when the thickness of OSC films is very low (<5 nm). (*The commonly used ionization gauge can be regarded as a kind of thermal plasma source, and usually works at the pressure of < 1 Pa. We found that the slight change of subthreshold voltage of OFETs in vacuum is the result of ionization gauge. For details, see our reply for your comment 11*).

Secondly, pressure and power both affect the plasma energy and density. At the same power, the lower pressure yields plasma with lower density but higher energy, which is prone to make a destructive effect on the OSC films. Therefore, the relative low power and high pressure are used in this work, as shown in

Table 1. In extreme cases, the power > 200 W cause macroscopic etching within 1 min or the pressure < 10 Pa leads to irreversible performance degradation of OFETs. Regarding the processing time, it is usually less than 1min. For the case of the ultrathin films (< 5 nm), 10 s is enough. For the case of the thicker films (> 30 nm), we found that extending the processing time is a better option than increasing the power. The voltage at the anode/cathode varies with the set power and pressure. It is usually hundreds of volts with the current of tens of amperes during the plasma treatment. The plasma is not uniform in our home-built system and increases in density as the distance with electrode is closer. In this work, the samples are fixed 10 cm away from the electrode, which is the edge of the plasma glow under the power of 50 W.

Thirdly, we found that H₂ plasma is effective than Ar and N₂ under the same process parameters. We propose two possible interpretations. ① Hydrogen ion has the smaller diameter than Ar and N, which may facilitate the diffusion or penetration of the plasma. ② The reducibility of hydrogen allows it to react with oxygen to achieve a better deoxygenation effect.

(2) The details of the light in re-doping. Firstly, we have fully discussed the role of light in the re-doping process in Supplementary Section 12: proposed de-doping and re-doping routes. As shown in **Fig. R4**, the role of the light is to excite an electron from the HOMO to the LUMO, and thus make it energetic enough to be transferred to a nearby interstitial oxygen molecule. Secondly, photooxidation does not occur in most cases. The illumination used in this experiment is a 15 W LED lamp from the viewing window. And the power density on the samples is measured to be **only 1.8 $\mu\text{W cm}^{-2}$** . OSCs are difficult to be oxidized in such low light. Even the easily oxidized molecule, P3HT, shows significant photooxidation under the light up to 100 mW cm^{-2} for several hours (*Adv. Energy Mater.* **2012**, 2, 1351-1357).

Figure R4. The role of light in the re-doping process.

Our revision: According to your comments, Table 1 and the discussion are added into Supplementary Section 5.

Your comments 3: Fig. 2 → What is the gas used for these experiments? Mention it in the caption, otherwise, the reader has to search for this information first.

Our reply: Thanks for pointing out this unclear point! The gas used for the experiments in Fig. 2 is H₂ because H₂ plasma show more efficiently under the same process parameters, as discussed in our reply for your comment 2.

Our revision: We have added this information into the caption of Fig. 2.

Your comments 4: How do you know that the donor-like traps exist in the OSC beforehand and are not created by the plasma? As you are showing the effect of the "plasma-annealing" for several materials, would the authors agree with the generalization that these donor-like traps exist in the majority of OSCs? What could be their origin?

Our reply: Thank you for your professional comments! We were also skeptical at first that the plasma would introduce defects or traps into OSCs, but the experimental results dispelled our concerns. As shown in **Fig. 3 b** and **c**, the performance of the re-doped OFET can almost basically recovered as before, even after successive de-doping and re-doping processes. You know that electrical performance of OFETs is very sensitive to defects or traps, far more than other spectral or morphological characterization, *e.g.*, IR, Raman and AFM. Therefore, it is the strongest evidence that our plasma treatment does not create new defects or traps.

Furthermore, it has been reported that the donor-like traps widely exist in many p-type OSCs even in the single crystal and show a similar exponential or gauss distribution in the energy gap. Their causes are believed to be impurities, structural defects and interface polar group etc. For example, Henning group reported that H₂O adsorbed in OSCs acts as a hole trap and significantly degrades the performance and stability of the OFETs (*Nat. Mater.* **2017**, 16, 356–362). Peter *et al.* investigated the distribution density of traps in the energy band of diF-TES ADT with bottom SiO₂ dielectric and top Cytop dielectric and found that the non-polar Cytop sample shows a remarkable reduction of the trap density of states (*Appl. Phys. Lett.* **2015**, 107, 103303).

Your comments 5: "Purified H₂, N₂, and Ar gases (99.999%) for de-doping treatment and further purified ..." Is this additional purification needed? Any indication for that?

Our reply: Thank you for your meticulous comment! In our lab, all gases used in this work (including O₂ and H₂O) enter the home-made system through a soft polyurethane (PU) tube. To avoid the cross contamination in the PU tube, we installed a deoxidize tube at the end of the PU tube when H₂, N₂ and Ar are used.

Our revision: For clarity, we added this explain in Method.

Your comments 6: I would appreciate it if the authors could add a summarizing table to the SI including all mobility values & threshold voltages for all materials in the doped and undoped states.

Our reply: Thank you very much for your kind and useful suggestion! First of all, we apologize that the mobility extraction for p-type OSCs in the de-doped state is incorrect. As you said in your comment 9, the large overdrive voltage should be used. Therefore, we re-extracted the mobility for most de-doped p-type OSCs. The samples are treated by 50 W H₂ plasma for 1 min. The results are shown in **Table 2**. As expected, the decrease in mobility of the most p-type OSCs is not very significant (detailed discussion see our reply for your comment 9). The mobility of rubrene drastically decreased, which may be due to the air gate structure of the rubrene transistor. The n-type materials generally exhibit remarkable increase in mobility with the improved threshold voltages, suggesting that oxygen has a very severe suppress of electron transport.

Table 2. Mobility and threshold voltage statistic for several organic semiconductors in the doped and de-doped states.

Material	P-type							N-type				
	DNTT	Rubrene	C ₈ -BTBT	CuPc	DPA	Pentacene	P3HT	PDI-C ₈	DHF-4T	N2200	PDI-CN	NDI-Cy6
μ_{doped} (cm ² V ⁻¹ s ⁻¹)	1.1	3.9	2.9	0.03	1.9	0.28	0.09	0.05	0.04	0.07	0.1	0.07
$\mu_{\text{de-doped}}$	0.6	0.03	1.2	0.02	1.1	0.1	0.04	2.2	0.65	0.78	0.9	0.8
$V_{\text{th doped}}$ (V)	-3	4	-9	2	-7	-20	17	16	28	28	9	20
$V_{\text{th de-doped}}$	-90	-10	-39	-70	-95	-100	-42	9	5	1	4.5	28*

*The transfer curve of NDI-Cy6 exhibit a nonlinear behavior after de-doping.

Our revision: We added Table 2 and the corresponding discussion into Supplementary Section 14.

Your comments 7: In order to understand the impact of the plasma treatment on industrial applications, it would be very valuable to know how long it takes for the redoping process to disappear under state-of-the-art glass-glass encapsulation conditions. For example, de-doped devices might show a much stronger bias stress due to rapid redoping. if this is the case, then the "plasma-annealing" might not be a desirable process as doped devices will not undergo this transition.

Our reply: Thank you for your professional and constructive suggestion! According to your comment, we assess the speed of the re-doping process under the glass-glass encapsulation condition. We encapsulate the de-doped DNTT OFET with a 1*1 cm² glass in a N₂ glove box and monitored its electrical performance for the next 14 days. As shown in **Fig. R5**, the performance recovery is very slow and limited for 14 days, but it is almost completely recovered when the encapsulation is broken and exposed to light. It indicates that de-doping devices are stable under encapsulation conditions. This result is also consistent with our measurements in inert gases conditions (**Fig. 3c**).

Figure R5. Performance recovery of the DNTT OFET from de-doping state under encapsulation condition.

Our revision: The corresponding data and discussion are added into Page 9 of main text and Supplementary Section 11. The descriptions of mobility decrease in main text and Supplementary Section 7 are deleted.

Your comments 8: Geometry details for the transistors & conductivity measurements should be reported (W/L etc.)

Our reply: Thanks for pointing out the missing details. The channel width/length for the thin-film transistors and conductivity measurements in this work is 5:1. For the cases of crystal transistors, *e.g.*, Rubrene and C₈-BTBT, the W/L is 2:1.

Our revision: We have added channel width and length into Method and double-checked the other experimental details.

Your comments 9: The mobility extracting for the de-doped hole transport FETs is in my opinion not valid as you are not comparing the devices at the same overdrive voltage. For example, a good measurement of the mobility for de-doped devices should be done *e.g.*, at -100V V_{GS}

Our reply: Thank you very much for pointing out this error! According to your comment, we re-measured the transfer curves of the de-doped OFETs under higher gate voltage, and extracted the valid mobility. As shown in **Fig. R6**, after plasma treatment, the hysteresis of curves obviously increases and non-linear occurs under high overdrive voltage, which indicates the increase of density of trap states. It is because the pre-empting effect of oxygen doping is removed by plasma treatment. Although the threshold voltage shifts significantly, no order of magnitude drop is in on-current observed under the similar overdrive voltage, suggesting no obvious change in mobility. In fact, the extracted mobility does show only a 50% decrease.

The slight decrease can be explained by the increase of trap density according to MTR model (*Phys. Rev. Lett.* **2004**, 00, 086602). This result is important. It indicates that mobility is a nature of materials and not significantly influenced by doping even considerable trap states inherent in OSCs, and that de-doping reduces the free charge density in semiconductors and results in higher drive voltage.

Figure R6. Transfer curves in linear region of the DNTT OFET during de-doping process. $V_{ds} = -5$ V.

Our revision: We added this data and discussion into Page 6 of main text and Supplementary Section 7.

Your comments 10: Fig 3b: 'switched from 0 to -30 V at 1 s intervals' - why do you see this continuous decrease of current with time. Is that bias-stress.

Our reply: We guess the continuous decrease of current that you said is the area circled in red (*i.e.*, **area 4**) in **Fig. R7**. It is not the result of bias-stress, but the decrease of photocurrent. You can see that the current of the OFET increases rapidly and exceeds the initial value after illuminated in O_2 (**area 3**). The exceeding current is attributable to non-equilibrium photon-generated carriers. These extra carriers gradually decrease and disappear in a few minutes and the decline of the current gradually stops, which is the process shown in area 4.

Figure R7. Switching cycle of the ultrathin (about 5 nm) DNNT OFET in multiple de-doping and re-doping

Our revision: For clarity, we added this explanation into Page 9 of the main text.

Your comments 11: Do the authors have any confirmation that sublimation & vacuum deposition does not give O₂-free films. From my experiments I can say that a DNNT film in UHV is not conductive but it becomes conductive in a glovebox. Hence, sublimation is a suited deoxygenation technique to clean materials, however, it cannot be applied to films that have been already fabricated  that is the advantage of the plasma.

Our reply: Thanks for your professional comment! We have discussed the point that sublimation cannot completely remove oxygen in OSCs. The experimental data and literature reports can be seen in **our reply for your comment 1**. Here we would like to discuss the experiments of DNNT film in UHV and glove box. The ionization gauge is very commonly used in vacuum system and usually works at the pressure of < 1 Pa. **However, everyone has overlooked that the effect of the ionization gauge is as the same as a thermal plasma source.** The decrease in conductivity and the shift of threshold voltage of OFETs from atmosphere to vacuum are always attributed to the oxygen desorption by high vacuum. **In fact, if the ionization gauge is shut down, the conductivity or threshold voltage of OFETs show no change even under UHV conditions!** (PS: According to our experiments, ionization gauge can effectively generate plasma in the pressure range of 0.01-1 Pa because the gases are too thin at very low pressure. So, it can be found that the performance change generally occurs within a few minutes after the startup of molecular pump) Accordingly, the plasma caused by ionization gauge is the reason for de-doping of the chemical adsorbed oxygen, *i.e.*, O₂⁻, rather than vacuum.

Our revision: We have added the discussion of ionization gauge into Supplementary Section 16.

Your comments 12: Fig. S12: Plasma de-doping 'with the energy barrier to charge transfer overcome by the plasma'. Why is this not possible by heat or light, but only plasma? What is the reason for using H₂?

Shouldn't inert gases be favored?

Our reply: Thank you for your insightful comment. We have discussed that the heat is not a very efficient de-doping method (our reply for your comment 1) and that light leads to re-doping rather than de-doping (our reply for your comment 2). So the question now is why plasma is more efficient.

We believe that the bombardment of ions with electric charges is more likely to disrupt the Coulomb interaction of organic semiconductors with oxygen (*i.e.*, organic radical cation and $O_2^{\bullet-}$) than heating. Plasma not only collides with organic molecules to transfer energy efficiently but also transfer charges with organic molecules, while heating transfers energy by phonon vibration.

We found that H_2 plasma is effective than Ar and N_2 under the same process parameters. We propose two possible interpretations. ① Hydrogen ion has the smaller diameter than Ar and N, which may facilitate the diffusion or penetration of the plasma. ② The reducibility of hydrogen allows it to react with oxygen to achieve a better deoxygenation effect.

Of course, the mechanism behind plasma de-doping needs further investigation. We will point this out in the manuscript, to get the attention of researchers in plasma and molecular doping.

Our revision: We added the possible mechanisms and corresponding discussion into Supplementary Section 12.

Your comments 13: In my opinion, the reason for the unique characteristic of removing O_2 trace amounts by plasma is not well described. The main text completely lacks any such description and in the SI, it is only briefly mentioned. Why do other methods like thermal annealing/annealing under inert atmospheres/illumination of OSCs fail to accomplish the same (were they also tested)?

Our reply: Thank you for concerning about this point. We apologize for this missing. According to your comments 1 and 12, we have discussed the possible mechanisms behind plasma de-doping process in detail. We also pointed that the proposed mechanisms are reasonable deduction and explanations based on our experiments and literatures, and it needs to be verified by further experimental and theoretical studies.

We have investigated thermal annealing of DNTT OFETs under N_2 . The temperature is set as $100\text{ }^\circ\text{C}$ because high temperature could lead to rapid film wetting. As shown in **Fig. R8**, the performance improvement of DNTT OFET can be observed after annealing 20 min, which is attributed to crystallinity enhancement. However, long annealing time significantly decreases the performance. This decrease in performance is attributed to morphology change caused by the migration of organic molecules (*we deliberated on this point in Nat. Commun. 2022, 13, 1480*) and thus is irreversible even under illumination in O_2 .

Figure R8. (a) Transfer curves of DNTT OFET with different annealing conditions. Images of the DNTT films after annealing 20 min (b) and 1 h (c) by laser scanning confocal microscope.

Our revision: We have added Fig. R8 and the corresponding discussion into Supplementary Section 12.

Reply for Reviewer #2

Your general comments: The authors report on oxygen trace effects on organic semiconductors in terms of charge transport. The results are interesting and the implication would be versatile. However, a few issues should be addressed before the potential publication in this journal.

Our reply: Thank you very much for your highly and constructive comments, which help us improve our manuscript! We address all your concerns one-by-one as follows.

Your comments 1: Lack of stability study:

The authors mentioned at the beginning that doping is important to stability, and indeed, doping/de-doping can be observed in the variation of device stability. It is necessary to examine stability and to include the stability results.

Our reply: Thank you for your professional comment! According to this comment, we examine stability of the OFETs before and after de-doping. The bias test is the most representative method to assess the stability of devices. In this test, V_{ds} and V_{gs} are set to -60 V. As shown in **Fig. R9**, the fresh DNTT OFET show negligible decrease in current within 3000 s whereas the current rapidly decrease within 2000 s after de-doping treatment (plasma 10 W, 1 min under H_2 200 Pa). The strong bias stress indicates the significant increase in trap density, which is attributed to trace oxygen de-doping resulting in the disappear of trap pre-empt effect. This result demonstrate that trace doping is important for device stability.

Figure R9. Bias stress test of the DNTT OFET before and after de-doping treatment (plasma 10 W, 1 min under H_2 200 Pa)

Our revision: We added Fig. R9 and the corresponding discussion into Page 7 of main text and Supplementary Section 7.

Your comments 2: Difference in microscopic structure:

It should be noted that the interaction between oxygen and organic semiconductor might be quite different for

polymers and small molecules. It means, C8-BTBT and P3HT, for example, are rather different from a microscopic point of view. It's unlikely that the same rule applies to both. So, the differences should be identified to figure out why the corresponding doping/de-doping arises.

Our reply: Thanks for your constructive comment! Indeed, there are some differences in the re-doping process between polymers and small molecules. In our experiments, re-doping process of P3HT (<30 min) is much faster than that of small molecules. According to your comment, we think it may be attributed to the microstructure difference between polymers and small molecules; oxygen diffusion in highly crystalline small molecules could be slower than disordered polymer chains.

Our revisions: According to your comment, we discussed the difference between polymers and small molecules in re-doping process due to microstructure in Supplementary Section 9.

Your comments 3: Impact of light:

The authors hint at light's effect on de-doping and re-doping, but without experimental result. It is necessary to check the role of light playing in those processes and to see if it has any consequences for the stability of the doped/undoped states. Perhaps do some stability tests to clarify the underlying mechanism by avoiding the influence of photo-generated carriers.

Our reply: Thank you for your professional comment. We have clearly demonstrated that light is necessary for re-doping process of OSCs in **Fig. 3b (Fig. R10 a)**. You can see that the current of the fresh device is about 6 μA and decrease rapidly to ~ 0 μA after de-doping (**process 1 in Fig. R10 a**). Then, oxygen was filled into the chamber in **process 2** but the current show no change, which indicates that re-doping does not occur under only oxygen. When the light is on in **process 3**, the current rapidly returns to its initial value. The current exceeding the initial value is caused by photogenerated carriers, and gradually degrade to equilibrium state in **process 4**, which suggests that photogenerated carriers have no influence on oxygen re-doping. This experiment vividly demonstrates the indispensable role of light in re-doping process. Combined with the recovery of EPR signals of organic radical cations and O_2^- (**Fig. 3e and f**), the mechanism of light in re-doping process is as follows (**Fig. R10 b i.e.**, Fig. S12): light excites a molecule to its excited state, and then the electron in the excited state is energetic enough to be transferred to a nearby interstitial oxygen molecule. As a result, the neutral organic molecule is oxidized to an organic radical cation (ORC), and O_2 is reduced to a superoxide anion (O_2^-).

Figure R10. (a) Switching cycle of the ultrathin (about 5 nm) DNTT OFET in multiple de-doping and re-doping processes with plasma treatment in H₂, Ar, and N₂. (b) Proposed de-doping and re-doping routes.

Our revision: We added the discussion of the role of light in re-doping process into Page 9 of main text.

Your comments 4: Need for low-temperature measurements:

To gain better understanding of the oxygen's role in the context of charge transport, it is necessary to perform low-temperature measurement. Such data could provide additional insight into the interaction between oxygen and charge carriers in organic semiconductors.

Our reply: Thank you for this constructive comment! According to your comment, we performed the low-temperature measurements for the DNTT OFET before and after de-doping, respectively (**Fig. R11**). As the temperature decreases, the on-state current of the fresh device increases at first and then decreases with negligible threshold voltage shift while the de-doped DNTT OFET shows obvious degradation in both on-state current and threshold voltage. The mobilities of the fresh and de-doped OFETs suggest different temperature dependence; the former increases at first and then decreases with the decrease of temperature, according with the feature of MTR model (*Phys. Rev. Lett.* **2004**, 00, 086602), and the latter monotonically decreases, which is a typical hopping behavior. These results match exactly with our insight of oxygen pre-empty effect. Mobility is governed by the time that a carrier transports within extended band (τ) and spends within a trap (τ_t), *i.e.*, $\mu_{eff} = \mu_0[\tau/(\tau + \tau_t)]$. For the case of doped state, oxygen pre-empties the trap states. The OFET shows a band-like mobility due to $\tau \gg \tau_t$ in the relative high temperature range, but a thermal activated mobility because τ_t increases in the relative low temperature range. For the case of de-doped state, oxygen pre-emptying effect is removed. A lot of trap states make $\tau_t \gg \tau$ in whole temperature range, resulting in a typical hopping mobility.

Figure R11. Temperature-dependent measurements of the OFET before and after de-doping. Transfer curves of the fresh device (a) and de-doped device (b). c, Temperature-dependent mobilities of the fresh and de-doped devices.

Our revision: We added Fig. R11 and the corresponding discussion into Page 7 of main text and Supplementary Section 7.

Your comments 5: Consideration of film thickness:

In single crystals (including C8-BTBT and DNNT, often called crystalline systems), oxygen may be just absorbed on the first layer. Thus, it is necessary to investigate the impact of film thickness in considering factors such as channel exposure and device structure. The manuscript should include data or discussion regarding the effect of varying thickness on oxygen-related phenomena, especially for small molecules.

Our reply: Thank for your professional comment and suggestion. Film thickness does have an important impact on the de-doping and re-doping process. However, our experimental data have suggested that the electrical performance of both ultrathin DNNT films (~ 5 nm) and routine films (20-30 nm) remarkably changed after plasma de-doping (Fig. R12, *i.e.*, Fig. 2f and Fig. 3d). It indicates that oxygen absorption is not only on the first layer. Furthermore, the results of TOF-SISM (Fig. 2e) clearly demonstrated that oxygen distribution is throughout the whole film.

Figure R12. De-doping of ultrathin (a) and routine thickness (b) DNTT OFETs.

Of course, de-doping of thin DNTT films is much faster and easier than that of thick films. According to your comment, we investigated the effect of varying thickness of DNTT OFETs on this process. The 5 nm, 20 nm, and 100 nm DNTT films were used, and the de-doping conditions are set as 5 W, 1 min. As shown in **Fig. R133**, the transfer characteristics of 5 nm film completely disappeared, while 20 nm and 100 nm films exhibited the shift of threshold voltage and the shift decreased with the increase of thickness. This result suggests that the thickness of organic films significantly hinders the de-doping effect of plasma, which could be due to the penetration limitation of plasma in crystalline film.

Figure R13. De-doping process of varying thickness of DNTT OFETs.

Our revision: According to your comments, we added the experiment and discussed the influence of thickness on de-doping and re-doping process. These results are added into Supplementary Section 7.

Your comments 6: Need for presentation improvement:

The English is fine, but improvement is still needed. Some mistakes or improper statements should be revised. ‘Underlying’ should be ‘underlying’. ‘Saturated mobility’ is better to be ‘saturation mobility’ or ‘mobility in the saturation regime’. ‘In the forbidden band of OFETs’ is not correct. OFETs have not an exact forbidden band. ‘on/off rate’ is often called ‘on/off ratio’. The statement of ‘the reversible de-doping and re-doping methods are clean’ is not good.

Furthermore, usage of adjective and adverb should be conservative. There are too many in a sentence, e.g., the 2nd paragraph, page 11

Our reply: Thank you for pointing these detailed mistakes! We have corrected all of them and double-checked our manuscript.

Your summative comments: In summary, the manuscript presents interesting findings regarding the role of trace oxygen in organic semiconductors, but it requires further refinement to address the aforementioned concerns and provide a more comprehensive understanding.

Our reply: Many thanks for all your valuable, professional and constructive comments and suggestions! These comments really help us improve our manuscript. We hope our replies and revisions can well address all your concerns and provide a more comprehensive understanding. Thanks again.

Reply for Reviewer #3

Your general comments: In this manuscript the authors discuss the impact of trace levels of oxygen on the properties of organic field-effect transistors. They propose an efficient method for de-doping which they then use in conjunction with various p-type semiconductors to flip the conduction to n-type. The doping is reversible and the p-type conductivity is regained upon re-doping. Given the fact that even very small quantities of oxygen make a huge difference, it is most likely that many, if not most of the past measurements have been conducted on organic semiconductors that have been unintentionally doped with oxygen. This work provides new insights on the impact of oxygen, a ubiquitous dopant, and hence is relevant not only for transistor research, but all devices based on organic semiconductors, and especially those operating in air. I recommend publication after the following concerns have been addressed:

Our reply: Thank you very much for your high and professional comments on our work! Your kind and constructive suggestions help us improve our manuscript. We have addressed all your concerns one-by-one as follows.

Your comments 1: The de-doping process is critical to this work, but the authors provide vague details about it. It is very important that this method is described in more detail so that other groups are able to reproduce it.

Our reply: Thanks for pointing the unclear experimental details! You and reviewer #1 both pointed out this issue. For clarity, we would like to add the de-doping details with a technical sheet (**Table 1**).

	DC	RF	Thermal
Plasma	Low density and energy	High density and energy	Low efficiency,
Pressure Range	10-1000 Pa	10-100 Pa	< 1 Pa
Power	0-300 W	0-100 W	0-20 W
Atmosphere	H ₂ , Ar, N ₂ (H ₂ optimum efficiency, No difference for Ar and N ₂)		
Treatment Results	Ineffective (< 5 W, > 1000 Pa) Nondestructive (< 50 W, > 50 Pa) Destructive (> 50 W, < 50 Pa)	Destructive (all conditions)	Ineffective PS: It can be only used for ultrathin films

Firstly, we discuss the plasma source tested in this work, *i.e.*, DC plasma, RF plasma and thermal plasma (*e.g.*, we used a heated tungsten wire to ionize the thin gas in the vacuum chamber). DC source yields plasma with low density and energy, which is applicable for organic materials. RF source yields plasma with high density and energy. We had tried to use RF source at the power of 1 W to treat OFETs, but the electrical

performance degraded and cannot be recovered. Thermal source is so inefficient that it only causes a slight shift of subthreshold voltage. This effect can be significant when the thickness of OSC films is very low (< 5 nm).

Secondly, pressure and power both affect the plasma energy and density. At the same power, the lower pressure yields plasma with lower density but higher energy, which is prone to make a destructive effect on the OSC films. Therefore, the relative low power and high pressure are used in this work, as shown in **Table 1**. In extreme cases, the power > 200 W cause macroscopic etching within 1 min or the pressure < 10 Pa lead to irreversible performance degradation of OFETs. Regarding the processing time, it is usually less than 1min. For the case of the ultrathin films (< 5 nm), 10 s is enough. For the case of the thicker films (> 30 nm), we found that extending the processing time is a better option than increasing the power. The voltage at the anode/cathode varies with the set power and pressure. It is usually hundreds of volts with the current of tens of amperes during the plasma treatment. The plasma is not uniform in our home-built system and increases in density as the distance with electrode is closer. In this work, the samples are fixed 10 cm away from the electrode, which is the edge of the plasma glow under the power of 50 W.

Thirdly, we found that H_2 plasma is effective than Ar and N_2 under the same process parameters. We propose two possible interpretations. ① Hydrogen ion has the smaller diameter than Ar and N, which may facilitate the diffusion or penetration of the plasma. ② The reducibility of hydrogen allows it to react with oxygen to achieve a better deoxygenation effect.

Our revision: The table 1 and corresponding discussion are added into the Methods and Supplementary Section 5.

Your comments 2: There is quite a bit of prior work that discussed oxygen doping in organic semiconductors and that has not been mentioned. The intro can be misleading for a non-expert in organic electronics, who might draw the inaccurate conclusion that this is the first discovery of oxygen doping in organic semiconductors.

Our reply: Thank you for your valuable and kind comment. Our manuscript missed out on a necessary discussion of prior oxygen doping works. The effect of oxygen on organic semiconductors (OSCs) has been noted since the last century. Oxygen was believed to be a cause of traps in both p- and n-type OSCs due to its oxidizing ability. While other studies suggested that some polymers such as P3HT show increase in conductivity and positive shift in threshold voltage when exposed to air, indicating the increase of hole carriers by oxygen. Although oxygen doping caused by air exposure in OSCs has been found over 20 years, no one noticed that trace oxygen is inherent in purified OSCs even without any exposure. How this trace oxygen works on OSCs and their devices has never been revealed. This is exactly what we unraveled in this work.

Our revision: To avoid misleading the audience, we clearly mention that oxygen doping has been found and discussed over 20 years in Page 12 of main text.

Your comments 3: The discussion around Figure 4 is key to the entire work, but it is very short. More details about these measurements should be provided. I understand how a p-type semiconductor can become n-type, but how does an n-type become p-type upon de-doping?

Our reply: Thank you for pointing the missing of the measurement details. According to your comment, we have added a sufficient discussion of the polarity inversion. In fact, the way inverted n-type OSCs to p-type is compensation doping rather than de-doping, *i.e.*, introducing excess oxygen doping by annealing and illumination in O₂ atmosphere to compensate the inherent electrons in n-type OSCs. When electrons are completely depleted, residual oxygen will dope OSCs to yield holes, endowing them p-type behaviors. This is different from the polarity inversion of p-type OSCs, in which p-type OSCs exhibit their intrinsic electron transport capability by de-doping.

Our revision: According to your comment, we added a discussion of the polarity inversion into Page 12 of main text.

Your comments 4: Can the authors be more quantitative on the oxygen quantity that would be needed to obtain p-type transport? Is that quantity dependent on the material? How?

Our reply: Thank you for your constructive comment. We believe that the oxygen concentration that would be needed for p-type transport depends on trap density in OSCs and devices. For example, the freshly prepared single crystal OFETs generally show relatively ideal performance because the traps in single crystal are so few that the inherent oxygen is enough to pre-empty them. Whereas thin film OFETs often suggest large threshold voltage and hysteresis at the beginning, and their performance would be significantly improved after exposed in air for a few days. It is because that there are too many traps in these devices and more external oxygen is needed to empty them. Accordingly, this question is very complicated; it may be related to the molecular orbital coupling, crystal quality, device structure, interface conditions and so on.

Furthermore, we also notice that inherent oxygen concentration varies in different materials and seems to be related to the energy gap. For example, pentacene and P3HT with the smaller energy gap (~ 2 eV) exhibit more oxygen concentration ($\sim 10^{16}$ cm⁻³) than DPA and C₈-BTBT (~ 3 eV), which may be due to the closer LUMO of these molecular to the reduction potential of oxygen. This conclusion requires more material investigation, and the related study is working. In addition, the difference in the oxygen concentration could explain why pentacene and P3HT OFETs often exhibit positive threshold voltage and depletion model. It also demonstrates that inherent oxygen concentration is not equal to the quantity needed by the materials and devices.

Your comments 5: How does the contribution of doping on the channel properties compare with that on the contact resistance?

Our reply: Thank you for arising this professional question. According to our experimental data, the contact resistance increases by ~ 2 times after de-doping at $V_g - V_{th} = 50$ V, and the total resistance increases by ~ 4 times under the same overdrive voltage. It suggests that the contribution of doping on channel and contact resistance is roughly the same under the high overdrive voltage. However, the doping contribution on channel will be much greater than that on contact resistance under overdrive voltage. For example, the contact resistance increases by ~ 4.75 times after de-doping at $V_g - V_{th} = 32$ V, but the total resistance increases by ~ 280 times. It indicates that trace oxygen doping is more prominent under low density of gate-induced carriers.

Our revision: We add above discussion into Supplementary Section 7.

REVIEWERS' COMMENTS

Reviewer #1 (Remarks to the Author):

I appreciate the enormous amount of work the authors spent in order to answer my questions, including a literature survey and additional experiments. I find the paper very interesting and happy to recommend it for publication.

Reviewer #2 (Remarks to the Author):

The reviewer appreciates the author's efforts in addressing most of the raised concerns. However, the distinct impacts of polymers and small molecules remain unconvincing. Given the significant differences in their microscopic structures and energy band structures, it's challenging to apply the discussed doping and de-doping mechanisms uniformly across such diverse systems. It's recommended that the author either removes the section on polymers or provides additional evidence to support their claims.

Considering the real-world processes, the paper's focus should be phenomena related to grain boundaries and surfaces in small molecule system. The complexity in achieving sufficiently pure polymer systems makes it more challenging to study these processes in common polymer systems.

Aside from this, the paper is nearing a publishable standard.

Reviewer #3 (Remarks to the Author):

The authors have addressed my comments adequately and I recommend publication in current form.

Reply for Reviewer #2

Your comments: However, the distinct impacts of polymers and small molecules remain unconvincing. Given the significant differences in their microscopic structures and energy band structures, it's challenging to apply the discussed doping and de-doping mechanisms uniformly across such diverse systems. It's recommended that the author either removes the section on polymers or provides additional evidence to support their claims. Considering the real-world processes, the paper's focus should be phenomena related to grain boundaries and surfaces in small molecule system. The complexity in achieving sufficiently pure polymer systems makes it more challenging to study these processes in common polymer systems.

Our reply: Thank you very much for your professional comments! Indeed, our work focus on the small molecule system and the evidence for polymer system is not very sufficient. In addition, previous reports also pointed that multiple doping species, such as ion-pair, charge-transfer complex and diradical cations, could form in polymer system (*Acc. Chem. Res.* **2016**, 49, 370–378; *Chem. Rev.* **2016**, 116, 13714–13751). Therefore, we agree that the do-doping and de-doping mechanism in polymer system is more complicated and likely different from that in small molecule system. To make audience aware of this point, we underline the complexity of polymer system in doping and de-doping mechanism and its possible inconsistency with the conclusions for small molecule system (Page 9-10).

However, we don't want to remove the section on polymers. Because we believe that the similar phenomena of polymers and small molecules in the de-doping and re-doping processes may attract more interests of the audiences in organic electronics, and it could be helpful to promote the further study on polymer doping mechanisms. Thank you very much for your understanding and supporting!